# iCanClean Improves Independent Component Analysis of Mobile Brain Imaging with EEG

**DOI:** 10.3390/s23020928

**Published:** 2023-01-13

**Authors:** Colton B. Gonsisko, Daniel P. Ferris, Ryan J. Downey

**Affiliations:** J. Crayton Pruitt Family Department of Biomedical Engineering, University of Florida, Gainesville, FL 32611, USA

**Keywords:** mobile brain imaging, electroencephalography (EEG), motion artifact removal, independent component analysis (ICA)

## Abstract

Motion artifacts hinder source-level analysis of mobile electroencephalography (EEG) data using independent component analysis (ICA). iCanClean is a novel cleaning algorithm that uses reference noise recordings to remove noisy EEG subspaces, but it has not been formally tested in a parameter sweep. The goal of this study was to test iCanClean’s ability to improve the ICA decomposition of EEG data corrupted by walking motion artifacts. Our primary objective was to determine optimal settings and performance in a parameter sweep (varying the window length and r^2^ cleaning aggressiveness). High-density EEG was recorded with 120 + 120 (dual-layer) EEG electrodes in young adults, high-functioning older adults, and low-functioning older adults. EEG data were decomposed by ICA after basic preprocessing and iCanClean. Components well-localized as dipoles (residual variance < 15%) and with high brain probability (ICLabel > 50%) were marked as ‘good’. We determined iCanClean’s optimal window length and cleaning aggressiveness to be 4-s and r^2^ = 0.65 for our data. At these settings, iCanClean improved the average number of good components from 8.4 to 13.2 (+57%). Good performance could be maintained with reduced sets of noise channels (12.7, 12.2, and 12.0 good components for 64, 32, and 16 noise channels, respectively). Overall, iCanClean shows promise as an effective method to clean mobile EEG data.

## 1. Introduction

Electroencephalography (EEG) is a promising tool for studying brain activity during whole body movement [1,2]. EEG noninvasively records electrocortical dynamics with high temporal resolution (250+ Hz), and high-density EEG caps (64+ channels) provide adequate spatial resolution (approximately 1 cm). Light-weight and portable EEG systems are now available on the market, allowing researchers to image the brain during a wide variety of activities [3]. However, there are several technical challenges to consider such as motion artifacts and source separation.

EEG signals measured at the scalp contain mixtures of many neural sources due to volume conduction [4]. Therefore, channel-level recordings are not independent but highly correlated with each other. EEG electrodes located directly over a neural area of interest record a relatively stronger signal than distantly placed electrodes, but they still record mixtures of multiple sources. Additionally, deep neural sources cannot be easily captured by direct, channel-level recordings since deep sources have no nearby electrodes. Thankfully, EEG analysis can be significantly improved by properly unmixing the EEG data and then reanalyzing the data at the source level, as opposed to the channel level. 

One popular and effective method to identify independent brain sources from mixed data is independent component analysis (ICA) [2,5]. ICA is a blind source separation method which linearly decomposes multiple channels of mixed data into independent components (ICs) by detecting linear subspaces of the EEG data which are maximally independent from each other based on higher-order statistics. Details regarding ICA and its computation can be found in [6,7,8]. ICA has been demonstrated to be a useful tool in isolating independent neural components which are highly dipolar [9]. There are many different implementations of ICA, with the most popular including Infomax [8], FastICA [10], and adaptive mixtures ICA (AMICA) [11]. A recent comparison study determined that AMICA performed the best based on multiple assessment variables including mutual information reduction in components and decomposition ‘dipolarity’ [9]. After appropriately unmixing the data into independent sources with ICA, the location of each source can be estimated by its topographic projection onto the EEG channels, which have known spatial location relative to the head. Specifically, ICA sources are often localized according to the best-fit dipole model [9,12], with low residual variance being a characteristic of clearer sources (topograph well explained by dipole model). Although low RV is useful for identifying brain sources, eye components and muscle components can also be well-fit by dipoles. Therefore, it can be useful to consider other information, such as a power spectral density, when determining if a particular component is likely a brain source. One popular option for auto-labeling components is ICLabel [13], which uses a convolutional neural network on crowd labeled training data to weigh labels and identify the probability that a component is from a given source. The possible sources from this algorithm are brain, muscle, eye, heart, line noise, channel noise, and other. ICLabel is built with a rich dataset of over 8000 components with over 30,000 labels and has been validated with expert labels.

In mobile data collections, EEG recordings have increased motion artifacts relative to stationary collections [14,15]. Motion artifacts can hinder ICA’s ability to decompose mixed EEG data into neural sources and subsequently localize them with dipole fitting and label them. For example, the scalp voltage can contain muscle (EMG) contamination and movement artifacts from whole body movement [15,16]. These artifacts can sometimes be identified by ICA [17,18], but often require removal prior to ICA. With large motion artifacts, there is a risk that ICA will not be able to extract high-quality brain components from the mixed data. Popular approaches for removing artifacts prior to ICA include adaptive filtering [19], principal component analysis [20,21], and wavelets [22]. Please see Seok et al. 2021 [23] for an in depth review of motion artifact removal methods for EEG.

We recently introduced a new method called iCanClean which uses canonical correlation analysis (CCA) and reference noise signals to detect and reject noise components [24]. With dual-layer EEG [25,26,27], noise electrodes are mechanically coupled to traditional (cortical) EEG electrodes to provide reference recordings of artifacts across space and time. By comparing the cortical electrode signals (recording mixtures of brain + noise) with noise electrodes (recording only mixtures of noise), iCanClean can remove noisy subspaces of EEG data without removing underlying brain signals. iCanClean should theoretically be able to improve the performance of ICA for source decomposition. Figure 1 summarizes the way in which motion artifacts can be removed from mobile EEG data using dual-layer EEG and the iCanClean algorithm. We provided recent evidence that iCanClean improves human data [28]; however, we did not perform a full parameter sweep.

The purposes of this study were to (1) determine whether iCanClean improves brain source estimation of mobile EEG data with ICA in multiple populations, and (2) determine the optimal settings for the two primary user-selectable parameters (window length and r^2^ in Figure 1) for maximally cleaning the data without removing electrocortical information. As the amount of noise can vary depending on testing conditions, the ability to improve brain source estimation is particularly important when processing large datasets with many different participants. Improved EEG cleaning methods should improve the fidelity of mobile brain imaging studies using high-density EEG for different types of real-world activities.

## 2. Materials and Methods

### 2.1. Data Collection

We collected mobile high-density EEG data from human participants walking over flat and uneven terrain on a custom treadmill [29], as depicted in Figure 2a. The data were collected as part of a larger study called Mind in Motion [30]. The EEG collection setup included a dual-layer cap, similar to the cap introduced in [25]. The dual-layer cap consisted of 120 scalp electrodes, as well as 120 outward-facing noise electrodes. The scalp electrodes and noise electrodes were mechanically fixed to each other (paired with a 3D printed plastic coupler), but electrically isolated from each other. Each participant walked on a treadmill: (1) at a fixed (personalized) speed with terrain of varying difficulty (Flat, Low, Medium, High) and (2) at varying speeds (0.25, 0.50, 0.75, and 1.00 ms^−1^) over flat terrain. Each participant walked for approximately 48 min. For this study, we analyzed data from 45 participants, including 15 from each of three groups: young adults (YA), high-functioning older adults (HFOA), and low-functioning older adults (LFOA). Young adults were between 20 and 40 years old, and older adults were 65+ years old. We determined lower extremity functional status amongst older individuals using the Short Physical Performance Battery [31]. There were three components to the test: walking speed over a short distance, the time to rise from a chair, and the ability to balance during various stances. Total scores ranged from 0 to 12. Individuals with scores < 10 out of 12 were considered to be low-functioning.

### 2.2. Data Processing

We processed data using custom written MATLAB scripts and EEGLAB [32]. Figure 2b shows the pipeline for EEG data importing and processing. We high-pass filtered EEG with a 1 Hz cutoff and average re-referenced channels (EEG and noise separately referenced to their own average). Since some channels were expected to contain no useful information (e.g., channels not properly gelled), we implemented a basic channel rejection method to remove large-amplitude channels. Channel amplitudes were quantified by taking the standard deviation across all samples (time points). Outlier channels with amplitudes greater than 3 times the median were removed. After this, all channels were re-referenced again, and this process was repeated for a second time. Across all 45 subjects, the average number of EEG channels that we rejected was 7.6 (out of 120), and the average number of noise channels that we rejected was 15.4 (out of 120).

Given the basic preprocessed data, we then performed a parameter sweep with iCanClean (Figure 2c). We varied two primary parameters of interest: the r^2^ threshold (cleaning aggressiveness) and the window length. As explained in [24], iCanClean removes components with a correlation greater than the given threshold, which can range from 0–1. Thus, the use of a higher r^2^ threshold (near 1) corresponds to less cleaning. For each parameter sweep, we sampled an r^2^ from 0.05 to 1 at increments of 0.05. Regarding the window length, we used four cases for this parameter: 1 s, 2 s, 4 s, and infinite. The 1, 2, and 4 s windows clean a small segment of data by comparing the local (in time) correlation between the cortical electrodes and the noise electrodes. Meanwhile, the infinite window uses the entire dataset (approximately 48 min long) when searching for correlation between cortical and noise electrodes. We also sought to demonstrate the iCanClean performance with fewer available noise channels. From the original data (120 noise channels), we downsampled to 64, 32, and 16 noise channels. This was carried out by considering the channel locations and retaining the most evenly spaced subset of noise channels using the *loc_subsets* function in EEGLAB. For each reduced dataset (64, 32, 16), a parameter sweep was conducted as before in the r^2^ range of [1, 0.05] with an increment of 0.05.

After cleaning the EEG channels with iCanClean, we performed source decomposition (Figure 2d). We used adaptive mixtures independent component analysis (AMICA) [11] to convert the channel data into independent components (candidate brain sources). Due to the high computational burden of repeated ICA calculations, we utilized a supercomputer cluster (University of Florida’s HiPerGator 3.0). We fit a dipole for each independent component using the dipfit plugin [32] with the generic boundary element head model and electrode locations. The result from dipole fitting is the best fit (most probable) location of the source along with the residual variance (RV) of the fit. Residual variance quantifies the quality of each independent component’s projection map (to the scalp sensors). Residual variance summarizes the mismatch between the observed (true) projection map and the projection map predicted by the best-fit electric dipole. If the residual variance is small, then the independent component is likely a physiological source such as brain or muscle. We then ran ICLabel, which is an EEGLAB plugin used to distinguish brain and non-brain sources based on an automated modeling framework that uses crowdsourced data [13]. ICLabel determines the probability of each independent component resembling brain data (as opposed to muscle activity, eye artifact, heart activity, line noise, channel noise, or other).

Finally, we evaluated the quality of the independent components based on the residual variance (RV) and ICLabel (Figure 2e). If a component had ≥50% probability of being brain according to ICLabel and an equivalent dipole could be fit with ≤15% residual variance, we identified it as a good component. The distribution of good components was plotted against the cleaning level to visually assess the effect of cleaning parameters on the number of good components.

### 2.3. Statistics

After determining the ideal cleaning parameters, we compared the number of good components for basic preprocessing (i.e., no iCanClean) with iCanClean at the ideal settings. We fit a mixed effects model to the data with the response variable being the number of good ICA components and the two predictor variables being the group (YA, HFOA, LFOA) and the iCanClean setting (off, ideal). We included first order terms as well as their interactions. We treated individual participants as a random effect. We tested all pairwise differences of the main effects and their interactions and corrected for multiple comparisons using Tukey’s simultaneous tests for differences of means. We were most interested in estimating iCanClean’s range of improvement, as opposed to simply testing whether there was any improvement using iCanClean. Rather than focusing on *p*-values, we also report 95% confidence intervals on the difference between iCanClean at ideal settings and without iCanClean. We constructed confidence intervals separately for each group (YA, HFOA, LFOA) and for all participants together.

## 3. Results

### 3.1. Parameter Sweeps

The performance across four windows (1 s, 2 s, 4 s, and infinite) is shown in Figure 3. For each plot, the number of good components is shown against the cleaning level. A horizontal dashed line is placed at the average value where the r^2^ cutoff was 1, showing the baseline performance (i.e., no iCanClean involved, only basic preprocessing). All 45 participants are included in each plot. 

The best window length tested was 4 s, followed by 2 s, 1 s, and infinite, in that order. For the infinite window, there was no increase from baseline. The maximum improvement for the 1 s window was 10.3 components from 8.4 (+1.9 components/+23% on average) at an r^2^ threshold of 0.95. The results are improved in the 2 s and 4 s windows across a wide cleaning range, from [0.95, 0.35]. For the 2 s window, the maximum increase in the average number of components compared to baseline was 12.3 from 8.5 (+3.9 components/+46%) at an r^2^ threshold of 0.85. Similarly, for the 4 s window, the maximum increase was 13.2 from 8.4 (+4.8 components/+57%) at an r^2^ threshold of 0.65. 

After determining the optimal iCanClean settings for our mobile EEG task, we performed a statistical analysis to compare basic preprocessing (no iCanClean) and best iCanClean (120 noise channels, 4 s window, r^2^ cutoff = 0.65). The mixed effects model fit the data well, with an adjusted R^2^ of 87.7%. There was a significant effect of participant group (F_2,42_ = 3.51, *p* = 0.039). There was also a significant effect of iCanClean (F_1,42_ = 60.14, *p* = 0.000). There were no significant interactions between iCanClean and the participant group (F_2,42_ = 1.61, *p* = 0.213). iCanClean resulted in an average improvement of 4.867 components (average across all 45 participants), with a 95% confidence interval of [3.600, 6.133].

We tested whether iCanClean worked similarly for all the participant groups. Shown in Figure 4 is the parameter sweep results for the 4 s window, separated by group. The shape of the plot for each group in Figure 4 is roughly the same as the grand average (across all participants) from Figure 3. From left to right across the plot, there is an increase in the number of physiological components over a broad range of r^2^ cutoffs until around 0.1. We were interested in characterizing the range of iCanClean’s performance for each group separately (no iCanClean versus the ideal settings of 4 s, r^2^ = 0.65, and 120 noise channels), so we used Tukey tests for differences of means to estimate confidence intervals for each group. YA, HFOA, and LFOA all showed a significant improvement with iCanClean (*p*-values = 0.001, 0.000, 0.022, respectively). On average, iCanClean improved YA by 4.67 components with a 95% confidence interval of [1.42, 7.91]. For the HFOA, there was an average improvement of 6.33 components with a confidence interval of [3.09, 9.58]. For the LFOA, the average improvement was 3.60 components with a confidence interval of [0.36, 6.84].

### 3.2. Reduced Channels

We also tested iCanClean’s performance with fewer reference noise channels available. Since the 4 s window generally performed best with the full set of noise sensors, we limited our tests with reduced noise sensors to the 4 s window. For the primary analysis, all 120 noise electrodes were used by iCanClean (100% available). For this secondary analysis, we tested results when using 64, 32, 16, and 0 (no cleaning). The results are shown in Figure 5. 

Based on Figure 5, ICA decomposition after using iCanClean did not have a drastic reduction in quality even when using relatively few noise channels. With only 16 noise channels (out of the original 120), iCanClean improved the average number of good components from 8.4 to 12.0 (+3.6/+43%). For both 32 and 64 noise channels, the number of high-quality components increased to 12.2 (+3.8/+45%) and 12.7 (+4.3/+51%), respectively. Although iCanClean yielded good results with a reduced number of noise electrodes, it performed better with more noise electrodes. The optimal r^2^ cutoff was affected by the number of noise channels. When fewer channels were used, the ideal r^2^ cutoff value was smaller (toward more aggressive settings). For example, when using 120 electrodes, the best cutoff was at 0.65. When only 16 channels were available, the optimal r^2^ cutoff decreased to 0.25. 

## 4. Discussion

The overall goal of this study was to test iCanClean’s ability to improve the ICA decomposition of EEG corrupted by walking motion artifacts. Specifically, we sought to evaluate iCanClean as a method to remove walking motion artifacts from human EEG data when using a dual-layer (EEG + noise) sensor system. We also wanted to provide guidance for optimal parameter settings when combining iCanClean with dual-layer EEG data. We determined that iCanClean was useful for removing motion artifacts and improved source-level analysis of mobile EEG data, as measured by the number of high-quality components resulting from independent component analysis. The optimal settings for our mobile EEG dataset were a 4 s window, an r^2^ cutoff of 0.65, and using the full set of 120 reference noise electrodes. Note that these findings were meant only to provide a starting point for others regarding ideal iCanClean settings. Ideal settings are likely to vary, for example, depending on the particular experimental design or hardware being used. 

These results provide guidance on the best window length choice for iCanClean to remove walking motion artifacts in the scenario where all 120 + 120 electrodes are available from the dual-layer EEG cap. We tested window lengths of 1 s, 2 s, 4 s, and infinite (infinite length meaning the full dataset, approximately 48 min long). We determined the 2 s and 4 s windows to be the most promising: the number of components increased by 3.9 (+46%) with a 2 s window and 4.8 (+57%) with the 4 s window. Note that although there was a general improvement as the window length increased (from 1 s to 2 s to 4 s), the full-length (infinite) window did not appear to improve the ICA decomposition outcome. This indicates that a moving window may be more appropriate for removing walking motion artifacts with iCanClean and dual-layer EEG. Given ongoing changes in walking dynamics, cap fit, electrode gel dispersion, and other time-varying parameters, data non-stationarity is likely to apply to most mobile human experiments with high-density EEG.

We also wanted to test whether iCanClean improved ICA decomposition of mobile EEG data from participants of different ages and lower extremity function levels. The 45 participants from this study were equally divided into three groups: 15 young adults (20 < age (years) < 40), 15 high functioning older adults (65 < age), and 15 low-functioning older adults (65 < age). We showed that with a 4 sec window and with the ideal r^2^ threshold (cleaning aggressiveness) of 0.65 from the grand average of all groups, the observed increases in the number of components were 4.7 (+42%) for YA, 6.4 (+93%) for HFOA, and 3.6 (+52%) for LFOA. The trend across the cleaning intensity (r^2^ cutoff) was similar for all three groups, and there was a wide range of cutoffs that improved data quality. Because there was a significant improvement with iCanClean in all three participant groups (and no interaction effects), we conclude that iCanClean is capable of improving mobile EEG data from a variety of populations.

Our last objective was to examine iCanClean’s ability to remove motion artifacts and improve ICA varied as a function of the number of reference noise channels available. The full number of available sensors was 120 EEG + 120 Noise electrodes. In addition to testing the full set of noise channels, we also tested iCanClean’s performance when it was limited to only having 64, 32, and 16 noise channels available. With fewer noise channels, there is a lesser degree of freedom for canonical correlation analysis to detect relationships between the cortical and noise channels. Therefore, we expected weaker correlations overall using fewer noise channels, and thus different optimal r^2^ cutoff values, depending on the number of noise channels. As expected, the optimal r^2^ cutoff decreased with the reduced number of noise channels (r^2^ = 0.65, 0.50, 0.35, 0.25 for 120, 64, 32, 16 channels, respectively). At these optimal cutoff values, we determined that iCanClean still performed well, even with a limited number of noise channels available. Compared to an improvement of 4.8 components (+57%) with all 120 noise sensors, we determined that iCanClean provided an average improvement of 4.3, 3.8, and 3.6 (+51%, +45%, +43%) when limited to 64, 32, and 16 noise channels, respectively. This suggests that iCanClean could still be useful to improve ICA results with a small number of noise electrodes available. 

This study builds upon previous work from our lab showing iCanClean could improve the ICA decomposition of table-tennis EEG data [28], a whole-body movement task prone to introducing motion artifacts. In [28], iCanClean was used with a 2 s window and an r^2^ cutoff of 0.85, and it was determined to improve the ICA decomposition from approximately 7 to 12 good components on average. The iCanClean parameters selected for the previous study were estimated by observation; a parameter sweep was not conducted. Here, we examined walking data, and we performed a detailed parameter sweep of 20 cleaning aggressiveness (r^2^) thresholds for 4 window lengths in 45 people. For our task (flat and uneven treadmill walking), we determined that a 2 s window was beneficial but the optimal window length was 4 s. We also determined that there were many suitable r^2^ thresholds, from 0.95 to 0.35 (best at 0.65). The previous study focused on young adults. Here, we tested older adults and low-functioning older adults in addition to young participants. Together, the current and recent studies demonstrate that iCanClean is a suitable tool for removing motion artifacts from human mobile EEG data for multiple whole-body movement tasks (walking, table-tennis) for multiple populations (younger, older, low-functioning).

There are some limitations to the present study. First, we did not attempt to remove muscle artifacts. Although dual-layer noise electrodes are good for collecting ambient motion sway or environmental noise, they are unable to capture muscle artifacts. For whole-body movement studies such as this, it is very likely that muscle artifacts are corrupting the data. The data collection setup for the Mind in Motion [30] study also obtained neck EMG data that could be incorporated into a future study of the algorithm. We focused on removing motion artifacts rather than neck muscle artifacts since motion artifacts are generally more problematic to ICA decomposition in our experience. ICLabel does have the ability to differentiate between muscle components and brain components. We determined that the number of muscle components also increases at the ideal iCanClean parameters, which is reasonable because the removal of motion artifacts will reveal both more brain and muscle components. Because the number of brain components increases when iCanClean is used, we can be confident that we are improving the data despite not removing muscle artifacts directly. Figure 6 depicts the number of muscle components according to ICLabel across different cleaning intensities with a 4 s window. The average number of muscle components is above 30, so effectively removing the muscle corruption will likely improve the number of brain components detected.

There are other popular ICA labeling algorithms in the literature we could have used, such as Multiple Artifact Rejection Algorithm (MARA) [33], Fully Automated Statistical Thresholding for EEG artifact Rejection (FASTER) [34], and SASICA [35]. However, these labeling algorithms attempt to label artifactual independent components for removal (IC pruning) rather than labeling of brain components (IC retention). For the purposes of this study, we wanted to quantify the improvement in brain-like components, not the decrease in artifactual-like components. For example, in preliminary work [36] we determined that MARA was an unsuitable metric for our parameter sweep as it favored overly aggressive cleaning. Specifically, MARA indicated that the ICA results always improved with more cleaning because MARA quantifies only the probability of a component containing an artifact, not the probability of a component being brain-like. Meanwhile, metrics such as ICLabel and ALICE [37] attempt to also label brain-like components. We decided to use ICLabel as our main component labeling method, combined with residual variance (from dipole fitting).

As another limitation, this study demonstrated the research utility of the iCanClean algorithm compared to basic preprocessing, but we did not directly compare the results with other algorithms, such as artifact subspace reconstruction [38]. This is due to the relatively high (computational) burden of running multiple sets of parameter sweeps for fair comparison. To process one data file, which contains approximately 48 min of high density dual-layer EEG data (120 + 120 channels, 500 Hz), approximately 33 min of computation time was required. Of that total time, roughly 5 min were spent on basic preprocessing (local computer), 3 min to run iCanClean (local computer), and approximately 25 min to perform ICA (supercomputer cluster utilizing 64 CPU cores). In total, we repeated this procedure approximately 6000 times, for a total of 3300 h (137.5 days) of computation time for this study. Along these lines, it is possible that iCanClean’s performance could be improved with further tweaks to the parameters, such as slightly lengthening the window from 4 s, but we had to limit our parameter search due to the computational cost of repeatedly running ICA on a supercomputer cluster.

Future work on iCanClean could build upon the findings from this parameter sweep and compare it with alternative cleaning methods to improve EEG source estimation for mobile brain/body imaging experiments. For example, one possible approach would be to combine different windowing methods in series, such as the 4 s window followed by the infinite window. Although the results indicate that the infinite window did not perform well alone, it could still be beneficial in removing stationary noise after a shorter window is used. The number of brain components after cleaning with other cleaning algorithms as compared with iCanClean is a logical extension of this work. This would provide important information regarding the situations in which different methods should be used. Finally, other recent component labeling algorithms such as ALICE [37] can be introduced when comparing different cleaning approaches to limit bias.

In conclusion, iCanClean is a promising, computationally efficient method to remove motion artifacts from mobile brain imaging data and improve subsequent source decomposition with ICA. iCanClean should facilitate mobile brain imaging research with EEG by allowing for a greater number of high-quality brain components to be detected and subsequently analyzed.

## Figures and Tables

**Figure 1 sensors-23-00928-f001:**
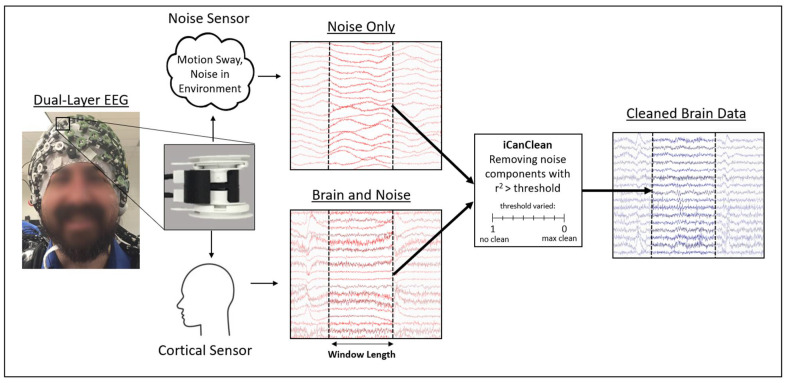
Illustration of the way iCanClean works. The participant is fitted with a dual-layer EEG cap consisting of inverted noise electrodes coupled to cortical electrodes. Example signals from dual-layer EEG are shown. The cortical electrodes record brain signals, but they also are commonly corrupted by noise and artifactual sources such as motion artifacts. The noise sensors are not in contact with the scalp and therefore only collect noise. iCanClean detects and removes subspaces of the EEG signals that are highly correlated with subspaces of the noise signals using canonical correlation analysis (CCA) [24]. The main tuning parameter for iCanClean is the r^2^ threshold. CCA components with an r^2^ correlation greater than a user-selected threshold are removed. The second tuning parameter for iCanClean is the length of the window over which CCA is calculated. Shorter windows could be useful for cleaning non-stationary data (e.g., large drifts), whereas longer windows provide better noise identification and removal, assuming the data are sufficiently stationary given the window length.

**Figure 2 sensors-23-00928-f002:**
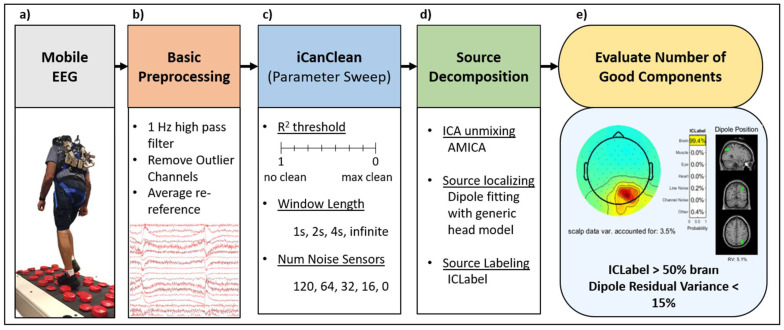
Processing pipeline from raw mobile EEG data to brain components. (**a**) Mobile EEG data were collected during flat and uneven terrain treadmill walking in younger, older, and low-functioning adults. (**b**) The data were high pass filtered and outlier channels were removed (channels with amplitudes greater than 3 × median). (**c**) iCanClean was used with different r^2^ thresholds (cleaning aggressiveness), time windows, and numbers of noise channels. (**d**) Brain sources were separated using adaptive mixture independent component analysis and they were labeled with ICLabel. (**e**) Components with >50% likelihood of being a brain source (from ICLabel) and <15% residual variance (from eeglab dipfit) were marked as good components.

**Figure 3 sensors-23-00928-f003:**
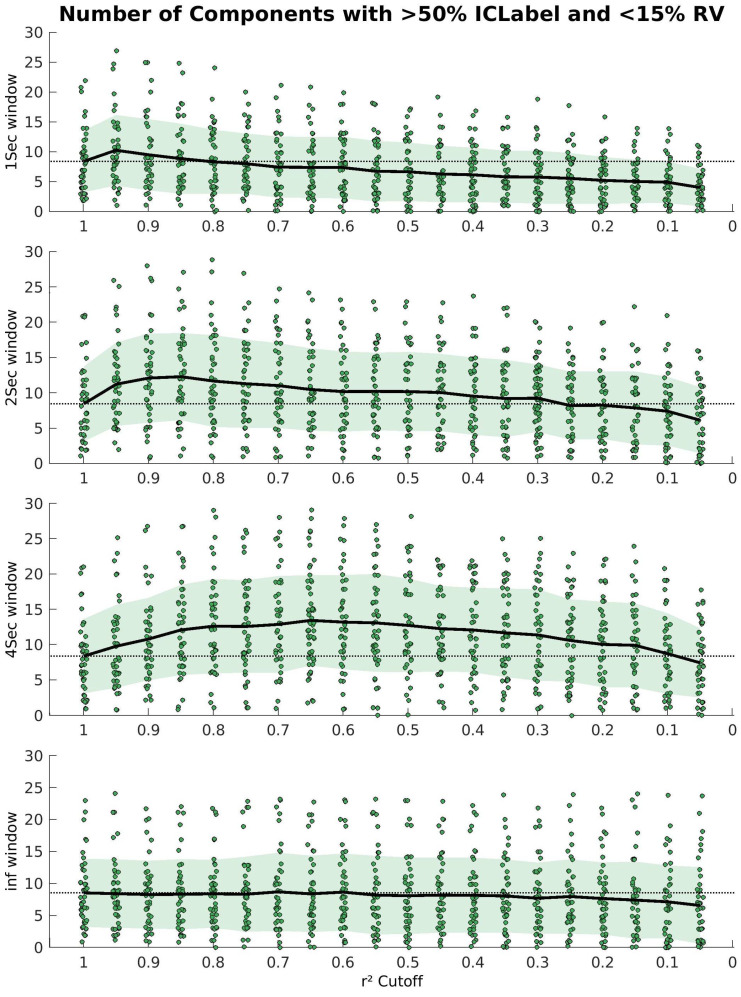
Results from parameter sweep for 1 s, 2 s, 4 s, and infinite windows. The number of high-quality components is plotted against the r^2^ cutoff. The average is shown as a thick black line, and the standard deviation is shaded in green. The baseline number of good components is shown as a dashed horizontal line in black. There is a visual increase in the performance in the 2 s and 4 s windows, but the 1 s and infinite windows do not increase much from baseline.

**Figure 4 sensors-23-00928-f004:**
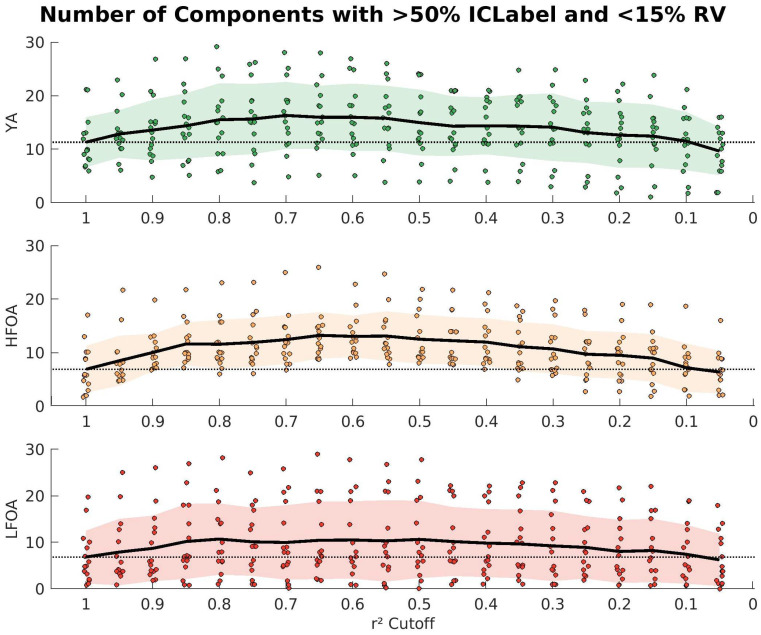
Results of the parameter sweep with separated subject groups for the 4 s window. Each plot shows a different group: young adults (green), high functioning older adults (yellow), and lower functioning older adults (red). The average from no cleaning is again shown as a dashed black line. The mean is a black line across the plot and the standard deviation is the shaded region.

**Figure 5 sensors-23-00928-f005:**
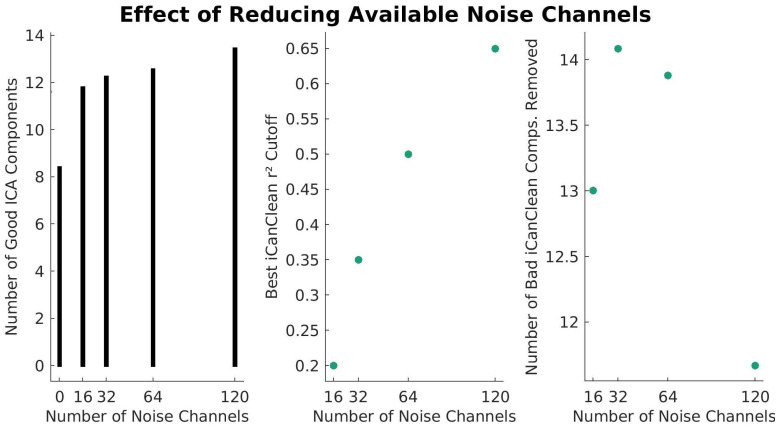
Results from iCanClean parameter sweep for 4 s window with reduced available noise channels. **Left**: the number of high-quality components is plotted against the number of noise channels used. **Middle**: the corresponding optimal r^2^ threshold (cleaning aggressiveness) is shown for each number of noise channels. **Right**: the number of bad iCanClean components removed as the number of noise recordings changes. Generally, as fewer noise channels are available, the r^2^ cutoff should be lowered (made more aggressive) to compensate, increasing the number of iCanClean components removed. Note that for 0 noise channels, iCanClean is not used, so there is no corresponding r^2^ cutoff.

**Figure 6 sensors-23-00928-f006:**
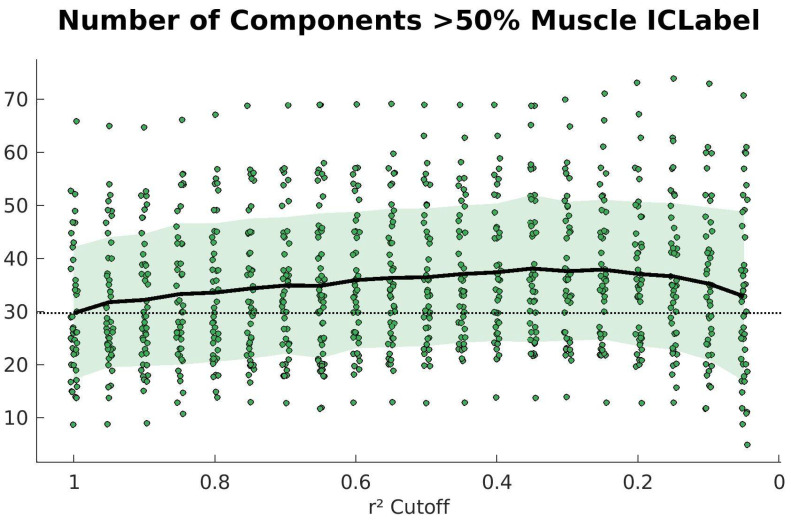
Number of muscle components labeled by ICLabel after using iCanClean to remove motion artifact (4 s window, range of cleaning intensities tested via r^2^ cutoffs). The average is shown as a bold black trace over the data points and the standard deviation is shaded in green. The average when iCanClean is not used is shown as a dotted black line. The number of muscle components increases at the ideal iCanClean parameter (r^2^ of 0.65). This increase is reasonable because the removal of motion artifacts should reveal both more brain and muscle components.

## Data Availability

The parameter sweep data is available in the Appendix A.

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
