# Peer review of "iCanClean Improves Independent Component Analysis of Mobile Brain Imaging with EEG"

_sensors, 2023, doi:10.3390/s23020928_

Round 1

Reviewer 1 Report

The Authors used a previously developed algorithm, iCanClean, for removing motion artifacts from EEG signals during walking, thus improving the performances of ICA in identifying brain sources. The work is methodologically sound and the results seem to be promising.
As a limitation of the study, the Authors did not try to remove muscle artifacts. However, it could be interesting to know whether such artifacts were identified by ICLabel during the source decomposition phase. If so, their impact on the results should have been negligible.

Reviewer 2 Report

1. Authors should emphasize contribution and novelty in the abstract. Moreover, the introduction needs to clarify the (1) motivation, (2) challenges, (3) contribution, (4) objectives, and (5) significance/implication.

2. It helps to appreciate the paper by having a related work section. The authors should consider more recent research done in the field of their study (especially in the years 2020 and 2021 onwards). The reader may want to see how this work differs from other previous works.

3. The authors should clearly describe related work in more detail, contrasting the limitations of the related works. Moreover, the reviewer recommend to ease the overview related works by using overview tables.

4. The article missed presenting the research novelty. In the sense that they do not highlight what is missing from each of the other proposals. The authors should provide enough proof to convince the reader of superiority of the proposed schemes over the existing works.

5. There is no discussion of user requirements, technological options and support for the decisions made at the design. The authors should include more technical details and explanations.

6. More experiments and some comparisons with other up-to-date methods should be addressed or added to back your claims to expand your experiments and analysis of results further.

7. It needs to highlight the research main contribution with some brief indications and numerical improvement percentages in section of result.

8. The conclusion and future work part can be extended to have a better understanding of the approach and issues related to that which can be taken into consideration for future work.

9. It is my understanding that a lot of works have been done in studying the related issues, such as [A] and [B]. The authors need to cite and discuss more current literature in the area.

[A] Seok, Dong Hee et al., Motion Artifact Removal Techniques for Wearable EEG and PPG Sensor Systems, Frontiers in Electronics, 2021.

[B] V. Roy, Effective EEG Artifact Removal from EEG Signal, DOI: 10.5772/intechopen.102698, 2022.

Reviewer 3 Report

I believe the paper address important problem of "clearning" EEG during motion and inspect one of the solution to this problem. The suggested method is very interesting and has already been previously introduced by this research group. Current ms. examined the optimal parameters for iCanClean - program for EEG cleaning of the dual-layer EEG recording. 

Through I support the publication of the ms. in the current form, I think it might be benificial to check what kind of additional ICA components reaserchers gain by using iCanClean with the parameters involved. The criteria for "good brain" component used in the ms. is clearly only one of possible approach to the problem.

I also sugest checking the results with other toolboxes for automatic ICA component labelling, e.g. recently introduced ALICE (http://alice.adase.org/, https://www.frontiersin.org/articles/10.3389/fninf.2021.720229/full) 

Round 2

Reviewer 2 Report

This paper has edited and revised according to the reviewer's suggestions.